# GENERALIZING CONSISTENCY MODELS FOR FAST IMAGE-TO-IMAGE TRANSLATION

## ABSTRACT

Diffusion models have shown strong performance in image-to-image (I2I) translation, combining high-fidelity generation with scalability to large-scale datasets. However, state-of-the-art models like Diffusion Bridge Models (DBMs) suffer from slow sampling speeds, requiring dozens to hundreds of expensive model evaluations. We address this limitation by extending Consistency Models (CMs), originally developed for noise-to-image (N2I) generation, to the I2I setting. We propose **Consistency Bridge Models** (CBMs), a new framework that enables few-step I2I translation from arbitrary source images without relying on pretrained diffusion models. CBMs inherit the efficiency of CMs while generalizing their theory to arbitrary non-Gaussian prior distributions. Evaluating on multiple datasets and image resolutions, we show that CBMs outperform prior work, reducing forward evaluations by up to 88%, and improving FID scores by up to 71%, offering an efficient framework for high-quality I2I translation.

## 1 INTRODUCTION

Diffusion Probabilistic Models (DPMs) have emerged as a powerful class of generative models that can produce highly realistic images through a gradual noise-to-data transformation, as illustrated in the Noise-to-Image (N2I) panel of Figure 2. This framework has driven major progress in image generation (Ho et al., 2020; Song et al., 2020b; Karras et al., 2022; Dhariwal & Nichol, 2021) and editing (Meng et al., 2021; Li et al., 2023; Liu et al., 2023; Su et al., 2022b).

However, sampling DPM is computationally intensive, often requiring hundreds of sequential steps for high-quality results. Advances by Song et al. (2020a); Lu et al. (2022a;b); Zhao et al. (2024) alleviate this cost, reducing sampling to around twenty steps while maintaining strong performance.

A related task, Image-to-Image (I2I) translation, converts images from one domain to another, *e.g.*, day to night, as illustrated in the I2I panel of Figure 2. Recently, Zhou et al. (2023) introduced

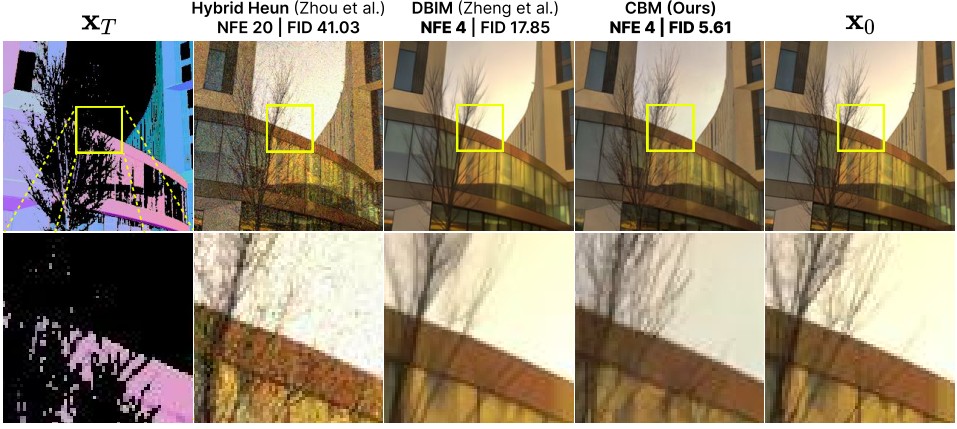

Figure 1: Few-step image synthesis (4 NFEs ↓) with high-quality generated details (5.61 FID ↓).

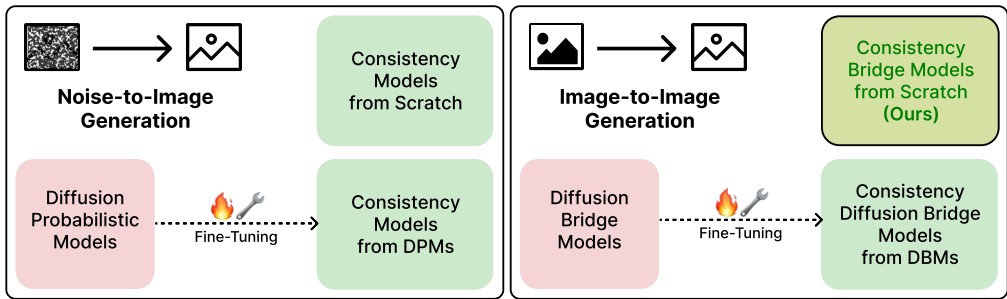

Figure 2: Overview of consistency-based generative modeling. **Left (Noise-to-Image Generation):** CMs (Song et al., 2023) generate images from noise, either trained from scratch or adapted from diffusion models. **Right (Image-to-Image Generation):** DBMs (Zhou et al., 2023) extend diffusion to I2I translation but need many steps. Our *Consistency Bridge Models* (CBMs) extend consistency theory to I2I and enable training from scratch, without fine-tuning (Geng et al., 2024; He et al., 2024).

Diffusion Bridge Models (DBMs): a generalization of the diffusion framework that constructs diffusion bridges between arbitrary image pairs, making high-quality diffusion-based I2I translation possible. While effective, DBMs also require dozens to hundreds of sampling steps for high-quality results, limiting their practicality. This raises a central question: *"Is it possible to design a framework for I2I translation that preserves quality without requiring dozens of costly sampling steps?"*

Consistency Models (CMs) (Song et al., 2023) improve efficiency by learning to *directly* map noise to data, enabling few-step N2I generation. Yet, their reliance on a Gaussian prior restricts them mainly to N2I generation, leaving I2I translation theoretically unsupported. To address this, we extend the consistency theory to I2I translation via Doob's $h$-transform (Doob, 1984). Unlike methods that finetune DBMs, our approach provides a principled framework for training consistency models for I2I translation. We introduce *Consistency Bridge Models* (CBMs): a family of standalone, trainable-from-scratch generative models for few-step, high-quality I2I translation.

CBMs significantly outperform their DBM counterparts, reducing the number of forward evaluations (NFEs) by up to $60\%$ while achieving superior Fréchet Inception Distance (FID) (Heusel et al., 2017) scores compared to DBM-based Hybrid Heun (Zhou et al., 2023) or DBIM (Zheng et al., 2024) (Section 4.1) sampling. Extensive experiments across multiple datasets and resolutions demonstrate the scalability, efficiency, and generalizability of CBMs, offering a faster and more cost-effective alternative to diffusion-based sampling. We will publicly release the code and model weights.

## 2 BACKGROUND AND RELATED WORK

### 2.1 NOISE-TO-IMAGE GENERATIVE MODELS

**Diffusion Probabilistic Models.** DPMs learn to generate images from a distribution $p_{\text{data}}(\mathbf{x})$ through a gradual denoising process (Ho et al., 2020; Song et al., 2020b; Karras et al., 2022; Dhariwal & Nichol, 2021). Starting with the prior Gaussian distribution $p_{\text{prior}}(\mathbf{x}) = \mathcal{N}(\mathbf{0}, \sigma_T^2 \mathbf{I})$ with mean $\mathbf{0}$ and standard deviation $\sigma_T > 0$, DPMs iteratively denoise $\mathbf{x}_T \sim p_{\text{prior}}(\mathbf{x})$ to recover the image sample $\mathbf{x}_0 \sim p_{\text{data}}(\mathbf{x})$. Surprisingly, the time evolution of this denoising process follows the Ordinary Differential Equation (ODE) (Anderson, 1982; Song et al., 2020b):

$$d\mathbf{x}_t = \left[ \boldsymbol{f}(\mathbf{x}_t, t) - \frac{1}{2} g(t)^2 \nabla_{\mathbf{x}_t} \log p_t(\mathbf{x}_t) \right] dt, \tag{1}$$

where time $t \in [0, T]$, $T > 0$, and $\boldsymbol{f}(\cdot, \cdot)$ and $g(\cdot)$ are known as the drift and diffusion coefficients, respectively. The perturbed images (*i.e.*, $\mathbf{x}_t$) are sampled from $p_t(\mathbf{x})$, with $\nabla_{\mathbf{x}_t} \log p_t(\mathbf{x})$ being its *score function*. This score function is learned by a neural network through the *Score Matching* objective (Hyvärinen, 2005; Song et al., 2020b). In addition, we can see that $p_0(\mathbf{x}) \equiv p_{\text{data}}(\mathbf{x})$ and $p_T(\mathbf{x}) \equiv p_{\text{prior}}(\mathbf{x})$. Song et al. (2020b) dub Equation (1) as the Probability Flow (PF) ODE. Note that specific hand-designed choices for $\boldsymbol{f}(\cdot, \cdot)$ and $g(\cdot)$ lead to *variance-preserving* (VP), *variance-exploding* (VE), or *TrigFlow* formulations (see Appendix A).

**Consistency Models.** Consistency models (Song et al., 2023; Song & Dhariwal, 2023) are generative models that learn to *directly* map an arbitrary point on the PF ODE trajectory (*i.e.*, $\mathbf{x}_t$, the perturbed image at time $t$) back to the initial point (*i.e.*, $\mathbf{x}_0$, the clean image). Song & Dhariwal (2023) showed that CMs trained from scratch can surpass the image generation quality of DPMs (Ho et al., 2020; Karras et al., 2022; Song et al., 2020a;b) in just a few steps. The noisy image $\mathbf{x}_t$ is obtained by perturbing the clean image: $\mathbf{x}_t = \mathbf{x}_0 + t\mathbf{z}$, where $\mathbf{z} \sim \mathcal{N}(\mathbf{0}, \mathbf{I})$. CMs are trained via the *Consistency Matching* objective:

$$\mathcal{L}_{\text{CM}}(\boldsymbol{\theta}) = \mathbb{E}\left[w(t,r)\, d(\boldsymbol{f}_{\boldsymbol{\theta}}(\mathbf{x}_t, t), \boldsymbol{f}_{\boldsymbol{\theta}}(\mathbf{x}_r, r))\right], \tag{2}$$

where $t$ and $r$ are successive time steps such that $0 \leq r < t \leq T$. After a consistency model $\boldsymbol{f}_{\boldsymbol{\theta}}$ is trained, images are generated via the Multistep Sampler (Song et al., 2023) in a few steps.

## 2.2 Image-to-Image Generative Models

**Diffusion Bridge Models.** Zhou et al. established a theoretical framework for image-to-image translation by leveraging Doob's h-transform (Doob, 1984; Rogers & Williams, 2000)–a mathematical tool enabling the construction of *stochastic bridges* between two arbitrary endpoints, $\mathbf{x}_0 \sim p_0(\mathbf{x})$ and $\mathbf{x}_T \sim p_T(\mathbf{x})$. This enables the conditioned forward diffusion from $\mathbf{x}_0$ to $\mathbf{x}_T$, whose time evolution is governed by the Stochastic Differential Equation (SDE):

$$\mathrm{d}\mathbf{x}_t = \left(\boldsymbol{f}(\mathbf{x}_t, t) - g(t)^2 \left[\nabla_{\mathbf{x}_t} \log p_t(\mathbf{x}_t \mid \mathbf{x}_T) - \nabla_{\mathbf{x}_t} \log p_t(\mathbf{x}_T \mid \mathbf{x}_t)\right]\right)\mathrm{d}t + g(t)\,\mathrm{d}\mathbf{w}_t, \tag{3}$$

where $\nabla_{\mathbf{x}_t} \log p_t(\mathbf{x}_t \mid \mathbf{x}_T)$ is the score of the *conditional probability* $p_t(\mathbf{x}_t \mid \mathbf{x}_T)$. In addition, Zhou et al. showed that Equation (3) has an ODE interpretation:

$$\frac{\mathrm{d}\mathbf{x}_t}{\mathrm{d}t} = \boldsymbol{f}(\mathbf{x}_t, t) - g(t)^2 \left(\frac{1}{2}\nabla_{\mathbf{x}_t} \log p_t(\mathbf{x}_t \mid \mathbf{x}_T) - \nabla_{\mathbf{x}_t} \log p_t(\mathbf{x}_T \mid \mathbf{x}_t)\right). \tag{4}$$

**Generating Images with DBMs.** Zhou et al. introduced the Hybrid Heun (HH) Sampler, which solves the SDE (Equation (3)) via the 1st-order Euler-Maruyama method, which is a simple numerical scheme for approximating SDEs (Maruyama, 1955), and the ODE (Equation (4)) via the 2nd-order Heun method (Heun, 1888; Süli & Mayers, 2003). However, this sampling procedure takes 119 NFEs, requiring multiple hours to generate images. Moreover, these images (showcased in Section 4) are low quality and contain noise even after the many NFEs. To alleviate this, Zheng et al. proposed DBIM, a Non-Markovian approach for sampling DBMs. While they achieved better results in fewer NFEs, they still require dozens of steps to achieve high-quality results, as shown in Section 4.

## 3 Generalizing Consistency Models for I2I Translation

As noted in Section 2.2, DBMs enable I2I translation but are computationally heavy; and while CMs are efficient, their theory is limited to N2I generation. To bridge this gap, we extend the consistency theory and introduce *Consistency Bridge Models* (CBMs): a generalization of consistency models for I2I translation achieved via Doob's *h*-transform. This allows CBMs to be capable of traversing from a known fixed endpoint back to a desired data point.

At the crux of CBMs lies the *consistency bridge function*, which maps any point $\mathbf{x}_t$ on the bridge trajectory back to the initial point $\mathbf{x}_0$ (illustrated in Figure 3). Furthermore, our CBMs can be regarded as a generalization of He et al. (2024)'s recent consistency-based approach named CDBMs, which we describe in Section 3.4.

Formally, let the solution trajectory to Equation (3) be given as $\{\mathbf{x}_t\}_{t \in [0,T]}$. We define the consistency bridge function as $\boldsymbol{f} : (\mathbf{x}_t, t, \mathbf{x}_T, T) \mapsto \mathbf{x}_0$, which satisfies the self-consistency property: $\boldsymbol{f}(\mathbf{x}_t, t, \mathbf{x}_T, T) = \boldsymbol{f}(\mathbf{x}_{t'}, t', \mathbf{x}_T, T) = \mathbf{x}_0$ for all $t, t' \in [0, T]$, with the boundary condition $\boldsymbol{f}(\mathbf{x}_0, 0, \mathbf{x}_T, T) = \mathbf{x}_0$. For brevity, we adopt $\boldsymbol{f}_{\boldsymbol{\theta}}(\mathbf{x}_t, t, \mathbf{x}_T) := \boldsymbol{f}_{\boldsymbol{\theta}}(\mathbf{x}_t, t, \mathbf{x}_T, T)$ in the text.

Our goal is to learn a neural network $\boldsymbol{f}_{\boldsymbol{\theta}}$ that estimates $\boldsymbol{f}$ by enforcing the self-consistency property, ensuring an effective Consistency Bridge Model. To the best of our knowledge, this is the first attempt to generalize the consistency framework to achieve I2I translation. We recap the design differences between DBMs, CMs, and CBMs in Table 1 before proceeding to the next section.

Table 1: Comparing CBMs to DBMs and CMs.

| Generative Models | Diffusion Bridge Models [37] | Consistency Models [28] | Consistency Bridge Models (Ours) |
|---|---|---|---|
| Theoretical Framework | Diffusion-based | Consistency-based | Consistency-based |
| Support for I2I Translation | Yes | No | Yes |
| Sampling Method(s) | Hybrid Heun [37], DBIM [36] | Multistep Sampler [28] | Algorithm 2 |

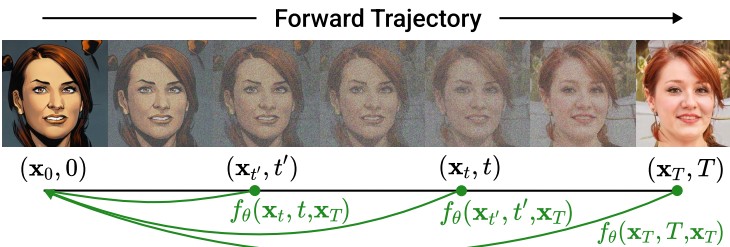

$$\text{Forward Trajectory} \longrightarrow$$

$(\mathbf{x}_0, 0)$  $(\mathbf{x}_{t'}, t')$  $(\mathbf{x}_t, t)$  $(\mathbf{x}_T, T)$

$f_\theta(\mathbf{x}_t, t, \mathbf{x}_T)$  $f_\theta(\mathbf{x}_{t'}, t', \mathbf{x}_T)$

$f_\theta(\mathbf{x}_T, T, \mathbf{x}_T)$

Figure 3: Consistency Bridge Models (CBM) for I2I Translation. The CBM $f_\theta$ learns to map any data point (*e.g.*, $\mathbf{x}_t$, $\mathbf{x}_{t'}$, and $\mathbf{x}_T$) on the Bridge trajectory back to the initial data point, $\mathbf{x}_0$.

## 3.1 PARAMETERIZING CBMS

For stability in training, we parametrize the CBM $f_\theta$ as a combination of input $\mathbf{x}_t$ and network $F_\theta$:

$$f_\theta(\mathbf{x}_t, t, \mathbf{x}_T, T) = c_{\text{skip}}(t, T)\,\mathbf{x}_t + c_{\text{out}}(t, T)\,F_\theta(c_{\text{in}}(t, T)\,\mathbf{x}_t, c_{\text{noise}}(t), \mathbf{x}_T),$$

where $c_{\text{in}}(t, T)$, $c_{\text{noise}}(t)$, $c_{\text{skip}}(t, T)$, and $c_{\text{out}}(t, T)$ are time-dependent differentiable scaling functions (Appendix B). Song et al. (2023) and Karras et al. (2022) also follow similar formulations to improve training stability. The input perturbed image at time $t$ is $\mathbf{x}_t \sim \mathcal{N}(\hat{\boldsymbol{\mu}}_t, \hat{\sigma}_t \mathbf{I})$, which can be rewritten as:

$$\mathbf{x}_t = \hat{\boldsymbol{\mu}}_t + \hat{\sigma}_t \mathbf{z}, \quad \mathbf{z} \sim \mathcal{N}(\mathbf{0}, \mathbf{I}), \tag{5}$$

where $\hat{\boldsymbol{\mu}}_t := \frac{\text{SNR}_T}{\text{SNR}_t}\frac{\alpha_t}{\alpha_T}\mathbf{x}_T + \alpha_t\left(1 - \frac{\text{SNR}_T}{\text{SNR}_t}\right)\mathbf{x}_0$, and $\hat{\sigma}_t^2 := \sigma_t^2\left(1 - \frac{\text{SNR}_T}{\text{SNR}_t}\right)$. $\alpha_t$ and $\sigma_t$ are functions of $t$, and similar to Song et al. (2023), we implement CBMs with the VE formulation (see Table 6).

## 3.2 TRAINING VIA THE CONSISTENCY BRIDGE MATCHING OBJECTIVE

Having described how to parametrize a CBM $f_\theta$, we now describe its training procedure for I2I Translation. Given two perturbed images $\mathbf{x}_{t_i}$ and $\mathbf{x}_{t_{i+1}}$ for time steps $0 \le t_i < t_{i+1} \le T$, we train $f_\theta$ by minimizing the *Consistency Bridge Matching Objective* over $\boldsymbol{\theta}$:

$$\mathcal{L}_{\text{CBM}}(\boldsymbol{\theta}) = \mathbb{E}\left[w(t_i, t_{i+1})\,d(f_\theta(\mathbf{x}_{t_{i+1}}, t_{i+1}, \mathbf{x}_T), f_{\theta^-}(\mathbf{x}_{t_i}, t_i, \mathbf{x}_T))\right], \tag{6}$$

where $w(\cdot, \cdot)$ is the weighting function, $d(\cdot, \cdot)$ is the distance function, $\boldsymbol{\theta}^-$ is the EMA of $\boldsymbol{\theta}$ (Song et al., 2023), and the expectation is taken over $t_i, t_{i+1} \sim p(t_i)$, $\mathbf{x}_0 \sim p_{\text{data}}(\mathbf{x})$, and $\mathbf{x}_T \sim p_{\text{prior}}(\mathbf{x})$.

As CBMs are generalizations of CMs, we can incorporate their training techniques for stable training, including the adoption of $w(t_i, t_{i+1}) = \frac{1}{t_{i+1} - t_i}$ as the weighting function and sampling the noise level $t_i$ from a lognormal distribution $i \sim n(i)$. These changes stabilize the training procedure of CMs, as observed by Song & Dhariwal (2023), which we find to be true for CBMs as well. Lastly, we adopt the Pseudo-Huber (Charbonnier et al., 1997) or Mean Squared Error as the distance function. By implementing these techniques in the training process, we reach Algorithm 1.

## 3.3 MULTISTEP BRIDGE SAMPLER: PERFORMING I2I TRANSLATION WITH CBMS

Given a prior image sample $\mathbf{x}_{t_N} = \mathbf{x}_T \sim p_{\text{prior}}(\mathbf{x})$ and $N$ sampling steps, we directly predict $\tilde{\mathbf{x}}_0$ from $\mathbf{x}_{t_N}$ using a well-trained CBM. We iteratively refine the output by computing the intermediate bridge points $\tilde{\mathbf{x}}_{t_n} \sim p_{t_n}(\mathbf{x}_{t_n} \mid \tilde{\mathbf{x}}_0, \mathbf{x}_T)$ as described in Equation (5), and predicting $\tilde{\mathbf{x}}_0$ from $\tilde{\mathbf{x}}_{t_n}$, for $t_n$ times, $t_n \in \{t_{N-1}, \cdots, t_2, t_1\}$. Intuitively, this iterative refinement step can be thought of as the 'predict-and-correct' or 'noise-and-denoise' step in diffusion-based samplers. This *Multistep Bridge (MB) Sampler* is capable of high-quality I2I translation. We frame it as Algorithm 2.

---

**Algorithm 1** Training Consistency Bridge Models

---

**Input:** Initial model parameter $\boldsymbol{\theta}$, initial distribution $p_0(\mathbf{x})$, prior distribution $p_\mathrm{T}(\mathbf{x})$, step discretization schedule $N(\cdot)$, lognormal distribution $n(i)$, distance metric $d(\cdot, \cdot)$, and learning rate $\eta$.
**Initialize:** Training iteration count, $k \leftarrow 0$
**repeat**
    Define noise discretization as a function of $k$: $T = t_{N(k)} > \cdots > t_2 > t_1$
    Sample $\mathbf{x}_0 \sim p_0(\mathbf{x})$, $\mathbf{x}_T \sim p_\mathrm{T}(\mathbf{x})$, and $i \sim n(i)$
    Sample $\mathbf{x}_{t_{i-1}} \sim p_{t_{i-1}}(\mathbf{x}_{t_{i-1}} \mid \mathbf{x}_T, \mathbf{x}_0)$, $\mathbf{x}_{t_i} \sim p_{t_i}(\mathbf{x}_{t_i} \mid \mathbf{x}_T, \mathbf{x}_0)$
    $\mathcal{L}_{\mathrm{CBM}}(\boldsymbol{\theta}) \leftarrow w(t_{i+1}, t_i)\, d(\boldsymbol{f_\theta}(\mathbf{x}_{t_{i+1}}, t_{i+1}, \mathbf{x}_T), \boldsymbol{f_{\theta^-}}(\mathbf{x}_{t_i}, t_i, \mathbf{x}_T))$     {Computing the Loss}
    $\boldsymbol{\theta} \leftarrow \boldsymbol{\theta} - \eta \nabla_{\boldsymbol{\theta}} \mathcal{L}_{\mathrm{CBM}}(\boldsymbol{\theta})$     {Updating the Model Parameters}
    $k \leftarrow k + 1$
**until** convergence

---

**Algorithm 2** Multistep Bridge (MB) Sampler for CBMs

---

**Inputs:** Pretrained CBM $\boldsymbol{f_\theta}(\cdot, \cdot, \cdot)$, Number of sampling steps $N$, Time steps $T = t_N > \cdots > t_1 > t_0 = 0$, and Prior distribution $p_T(\mathbf{x})$.

**Initial Step:** Sample $\tilde{\mathbf{x}}_{t_N} = \mathbf{x}_T \sim p_T(\mathbf{x})$, with $\mathrm{SNR}_T \leftarrow \alpha_T^2 / \sigma_T^2$

**Subsequent Stochastic Refinement:**
**for** $i = N$ **to** 1 **do**
    $\tilde{\mathbf{x}}_0 \leftarrow \boldsymbol{f_\theta}(\tilde{\mathbf{x}}_{t_i}, t_i, \mathbf{x}_T)$     {▷ Generate $\tilde{\mathbf{x}}_0$ from $\tilde{\mathbf{x}}_{t_i}$ using CBM $\boldsymbol{f_\theta}$}
    **if** $i > 1$ **then**
        Sample $\mathbf{z}_{i-1} \sim \mathcal{N}(\mathbf{0}, \boldsymbol{I})$, with $\mathrm{SNR}_{t_{i-1}} \leftarrow \alpha_{t_{i-1}}^2 / \sigma_{t_{i-1}}^2$

$$\tilde{\mathbf{x}}_{t_{i-1}} \leftarrow \frac{\alpha_{t_{i-1}}}{\alpha_T} \frac{\mathrm{SNR}_T}{\mathrm{SNR}_{t_{i-1}}} \mathbf{x}_T + \alpha_{t_{i-1}} \left(1 - \frac{\mathrm{SNR}_T}{\mathrm{SNR}_{t_{i-1}}}\right) \tilde{\mathbf{x}}_0 + \sigma_{t_{i-1}} \sqrt{\left(1 - \frac{\mathrm{SNR}_T}{\mathrm{SNR}_{t_{i-1}}}\right)} \mathbf{z}_{i-1}$$

    **end if**
**end for**
**Output:** $\tilde{\mathbf{x}}_0$     {▷ Final translated image}

---

### 3.4 RELATION TO PRIOR CONSISTENCY-BASED APPROACHES

For CMs, the perturbations $\mathbf{x}_t$ are simply computed as $\mathbf{x}_t = \mathbf{x}_0 + t\mathbf{z}$ (Section 2.1), consistent with N2I generation (Song et al., 2023). For CBMs, $\mathbf{x}_t$ is instead formed from $\mathbf{x}_0$, $\mathbf{x}_T$, and noise (Equation (5)), reflecting the paired I2I setting where intermediate states interpolate between prior and target images. Architecturally, CMs take $\mathbf{x}_t$ and time step $t$ as inputs, while CBMs additionally use $\mathbf{x}_T$ to guide translation via the consistency bridge function. During sampling, CMs refine Gaussian noise to $\tilde{\mathbf{x}}_0$, whereas CBMs translate from prior $\mathbf{x}_T$ to target $\tilde{\mathbf{x}}_0$ using our MB Sampler.

Meanwhile, He et al. (2024)'s CDBMs are closely related to CBMs. Instead of training from scratch, CDBMs simply progressively distill a pretrained DBM via the training procedure described by Geng et al. (2024), which uses the consistency bridge function's boundary condition (discussed in Section 3). While CDBMs can generate high-quality images, they are constrained by requiring a pretrained DBM that needs further finetuning, which is a major computational limitation. In contrast, CBMs require relatively much lower computational costs to train from scratch while producing high-quality images, as evidenced by Table 2 and Figure 6, while taking demonstrably lower training resources.

## 4 EXPERIMENTS

We train all CBMs entirely from scratch using the Consistency Bridge Matching objective (Section 3.2) and describe their training hyperparameters in Appendix C.1. Our evaluation spans across three datasets with varying resolutions: *Edges2Handbags* (E2H, $64 \times 64$) (Isola et al., 2017), *DIODE* ($256 \times 256$) (Vasiljevic et al., 2019), and *Face2Comics* ($256 \times 256$) (Sxela, 2021). Furthermore, we additionally evaluate on Face2Comics ($64 \times 64$) and ($128 \times 128$) resolutions to demonstrate the scalability of CBMs. On DIODE, the task is to translate dense surface normal maps into color images, while Face2Comics requires mapping natural human faces into stylized comic renderings.

We assess generation quality using FID (Heusel et al., 2017) and efficiency via the number of forward evaluations (NFEs) (Song et al., 2020a; Lu et al., 2022a), and report previously published results of Zhou et al. (2023) and Zheng et al. (2024). For baselines, we compare against diffusion-based I2I methods, including DDIB (Su et al., 2022a), SDEdit (Meng et al., 2021), Rectified Flow (Liu et al., 2022), I$^2$SB (Liu et al., 2023), and DDBM (Zhou et al., 2023), the current state-of-the-art.

For E2H and DIODE, we sample images using the pretrained DBM weights published by Zhou et al. with the Hybrid Heun and DBIM samplers. Lastly, as pretrained weights do not exist for Face2Comics and CelebAMask, we identically train both DBMs and CBMs from scratch for fairness.

### 4.1 RESULTS

**Image Translation on DIODE** $(256{\times}256)$ **and E2H** $(64{\times}64)$. We observe that CBMs demonstrate improvements in quality, speed, and computational efficiency compared to prior work based on GAN and diffusion. We compare CBMs to prior state-of-the-art works on both DIODE and E2H datasets and summarize it in Table 2. We observe that a DBM with the Hybrid Heun (HH) sampler can only reach 5.42 and 1.83 FIDs on DIODE and E2H respectively, even though given a very generous computation budget of 119 NFEs. Meanwhile, DBM with DBIM on the DIODE dataset achieves a FID of 17.85 with 4 NFEs, improving to 7.99 with an extended budget of 10 NFEs. A similar trend is observed on E2H, where DBIM reaches 4.14 and 2.49 FID with 4 and 10 NFEs, respectively. In contrast, our CBM (with the Multistep Bridge Sampler) significantly outperforms DBMs with HH and DBIM, achieving 3.79 FID on DIODE and 1.22 FID on E2H with just 6 NFEs.

CBMs achieve state-of-the-art performance with substantially reduced computational overhead, operating within a notably compact architectural design, as detailed in Appendix 9. By effectively balancing the trade-off between speed (fewer NFEs) and quality (lower FID), our CBMs enhance both aspects while requiring fewer training samples, leading to shorter training durations and fewer training iterations, as illustrated in Figure 6 on DIODE and E2H.

The qualitative results in Figures 4 and 5 align with the quantitative results, where we can observe that our CBM generates higher-fidelity images in fewer steps. In Figure 4, the images generated with HH still contain large amounts of noise at 20 NFEs, while our CBM achieves high-quality images with just 4 NFEs. Although CDBMs achieve a lower FID of 2.93 on DIODE and 0.80 on E2H, they require considerably more training and computational resources, as presented in Figure 6. Please refer to Section 3.4 for a discussion on why CDBM is not a suitable baseline comparison for CBMs.

**Label-to-Face Generation on CelebAMask-HQ** $(256{\times}256)$. Figure 7 and Table 3 illustrate the advantages of our CBM over prior methods in generating high-quality facial images from segmentation masks. At just 6 NFEs, CBM achieves an FID of 13.14, outperforming the 44.92 FID of

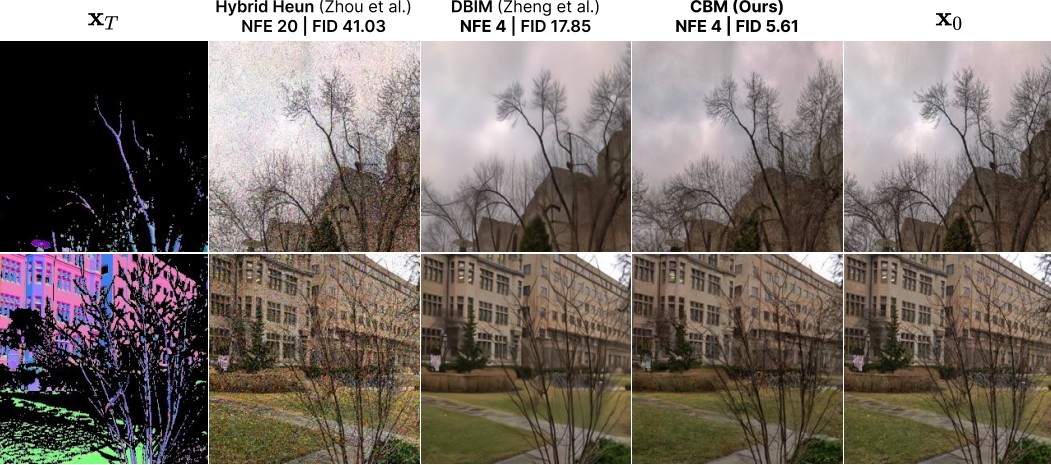

Figure 4: Qualitative comparison on DIODE $256 \times 256$ [33]. Our CBMs with Multistep Bridge Sampler generate higher-quality images with fewer NFEs compared to DBMs with HH (Zhou et al., 2023) and DBIM (Zheng et al., 2024) (Table 2). $\mathbf{x}_T$ are Surface Normal Maps, and $\mathbf{x}_0$ are Images.

Table 2: Quantitative performance comparison of various diffusion-based and consistency-based models on the DIODE [33] and Edges2Handbags (E2H) [12]. We report model size, training duration (in number of images), NFEs, and FID to assess efficiency and generation quality. Traditional diffusion methods (e.g., DDIB, SDEdit, Rectified Flow, I²SB) require significantly higher NFEs to achieve competitive FID scores. DBIM sampler generates higher-quality images but require large NFEs. While CDBMs achieve strong FID scores with minimal NFEs, they require substantial training costs. Our proposed Consistency Bridge Models (CBMs) outperform prior methods by achieving state-of-the-art FID scores with $\leq 6$ NFEs, reduced training cost, and similar or smaller model sizes.

| Method | DIODE ($256\times256$) [33] | | | | Edges2Handbags (E2H) ($64\times64$) [12] | | | |
|---|---|---|---|---|---|---|---|---|
| | Model Size ($\downarrow$) | Training Duration ($\downarrow$) | NFE ($\downarrow$) | FID ($\downarrow$) | Model Size ($\downarrow$) | Training Duration ($\downarrow$) | NFE ($\downarrow$) | FID ($\downarrow$) |
| *Other Diffusion-based Methods:* | | | | | | | | |
| DDIB [29] | – | – | $\geq 40$ | 242.30 | – | – | $\geq 40$ | 186.84 |
| SDEdit [23] | – | – | $\geq 40$ | 31.14 | – | – | $\geq 40$ | 26.50 |
| Rectified Flow [17] | – | – | $\geq 40$ | 25.30 | – | – | $\geq 40$ | 77.18 |
| I²SB [16] | – | – | $\geq 40$ | 9.34 | – | – | $\geq 40$ | 7.43 |
| *Diffusion Bridge Models:* | | | | | | | | |
| Hybrid Heun [37] | 534 Mil. | $25.6\times10^6$ | 20 | 41.03 | 284 Mil. | $102.4\times10^6$ | 20 | 46.74 |
| | 534 Mil. | $25.6\times10^6$ | 119 | 5.42 | 284 Mil. | $102.4\times10^6$ | 119 | 1.83 |
| DBIM [36] | 534 Mil. | $25.6\times10^6$ | 4 | 17.85 | 284 Mil. | $102.4\times10^6$ | 4 | 4.14 |
| | 534 Mil. | $25.6\times10^6$ | 10 | 7.99 | 284 Mil. | $102.4\times10^6$ | 10 | 2.49 |
| *Further Fine-Tuning of DBMs:* | | | | | | | | |
| CDBM† [7] | 534 Mil. | $33.3\times10^6$ | 2 | **2.93** | 284 Mil. | $110.1\times10^6$ | 2 | **0.80** |
| *Consistency-based Models (Training from Scratch):* | | | | | | | | |
| CBM (Ours) | 161 Mil. | $11.5\times10^6$ | 4 | 5.61 | 284 Mil. | $90.6\times10^6$ | 4 | 1.96 |
| | 161 Mil. | $11.5\times10^6$ | 6 | **3.79** | 284 Mil. | $90.6\times10^6$ | 6 | **1.22** |

† CDBMs require a pretrained DBM that is further finetuned (refer to Section 3.4 and Figure 6).

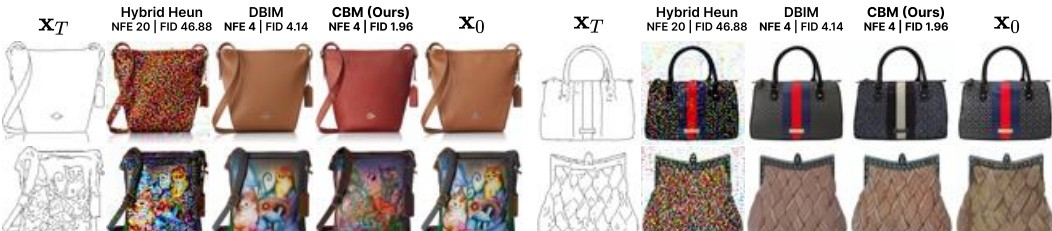

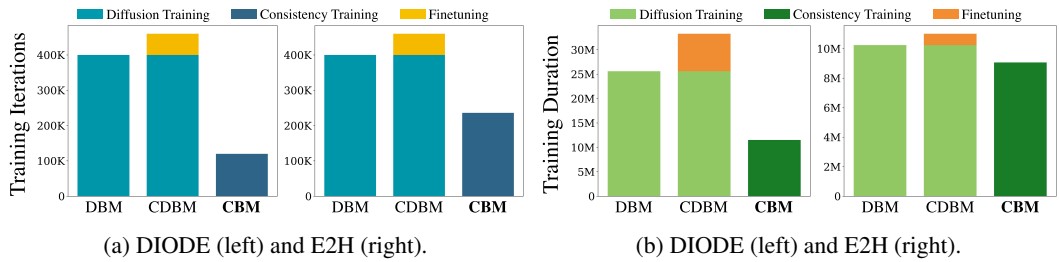

Figure 5: Qualitative comparison on E2H $64 \times 64$ (Isola et al., 2017). CBMs produce superior visual outputs with fewer NFEs relative to the HH (Zhou et al., 2023) and DBIM (Zheng et al., 2024) samplers for DBMs (Table 2). $\mathbf{x}_T$ are Edges, and $\mathbf{x}_0$ are Images.

(a) DIODE (left) and E2H (right).

(b) DIODE (left) and E2H (right).

Figure 6: Comparing **(a):** Number of Training Iterations, and **(b):** Training Duration (images). CBMs demonstrably require significantly less computational resource compared to DBMs and CDBMs, taking just $26\%$ and $51\%$ of the training iterations for DIODE and E2H, respectively.

DBIM and surpassing all GAN-based and diffusion baselines. While DBIM tends to produce overly smooth outputs, often lacking fine-grained details and exhibiting a plastic-like appearance, CBM preserves intricate facial structures and textures. This leads to sharper, more anatomically faithful generations that enhance both realism and alignment with $\mathbf{x}_T$.

**Image Stylization on Face2Comics ($256\times256$).** Similarly, as shown in Figure 8 and Table 4, CBMs outperform DBMs at both 7 and 14 NFEs, consistently generating higher-quality images. Analytical studies presented in the subsequent discussion further demonstrate that increasing the

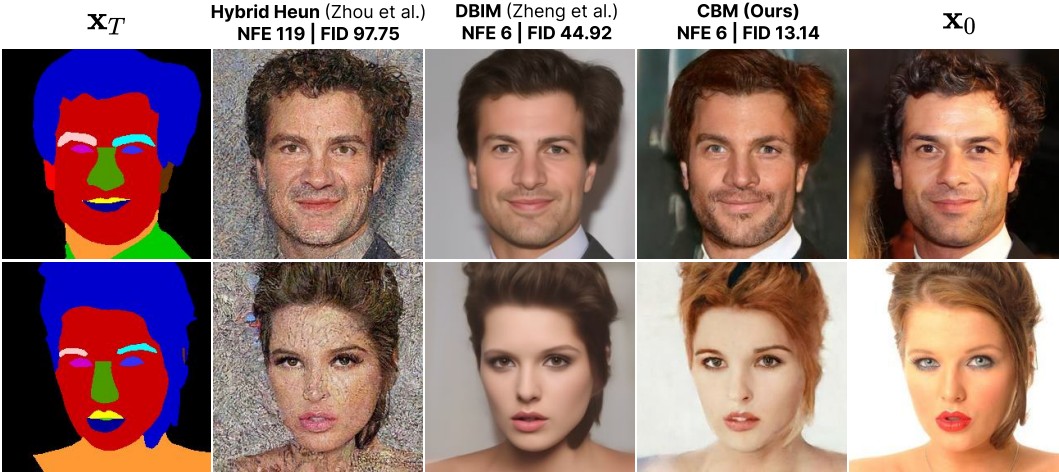

Figure 7: Qualitative comparison of CBMs with prior diffusion-based methods on the CelebAMask-HQ $256 \times 256$ dataset (Lee et al., 2020). CBMs produce sharper, more coherent, and visually realistic outputs, demonstrating superior semantic alignment and style fidelity compared to DBM baselines with the HH and DBIM samplers.

Table 5: Evaluating the scalability of CBMs compared to DBMs on increasing resolutions of the Face2Comics dataset (Sxela, 2021). We train a DBM and CBM with the same model design and scale it to resolutions of $64 \times 64$ and $128 \times 128$. Our CBM with the Multistep Bridge (MB) Sampler significantly outperforms DBMs with the Hybrid Heun and DBIM Samplers.

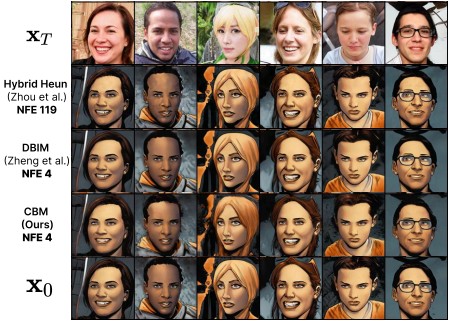

Figure 9: Qualitative comparison at $128 \times 128$ resolution on Face2Comics.

| Method | Face2Comics ($64 \times 64$) | | | Face2Comics ($128 \times 128$) | | |
|---|---|---|---|---|---|---|
| | Model Size | NFE ($\downarrow$) | FID ($\downarrow$) | Model Size | NFE ($\downarrow$) | FID ($\downarrow$) |
| *Diffusion Bridge Models:* | | | | | | |
| Hybrid Heun [37] | 101 Mil. | 119 | 2.50 | 139 Mil. | 119 | 31.28 |
| DBIM [36] | 101 Mil. | 6 | 2.78 | 139 Mil. | 6 | 21.76 |
| *Consistency Bridge Models:* | | | | | | |
| CBM (Ours) | 101 Mil. | 6 | **0.89** | 139 Mil. | 6 | **12.24** |

model capacity of CBMs leads to notable performance gains over prior approaches. As observed in Table 4, CBMs consistently achieve lower FID scores with far fewer NFEs across all resolutions compared to baseline DBMs. At a resolution of $256 \times 256$, our CBM attains an FID of 3.64 with just 6 NFEs, outperforming HH and DBIM samplers, which yield FIDs of 13.29 (in 6 NFEs) and 52.39 (in 20 NFEs), respectively. We observe similar trends at $64 \times 64$ and $128 \times 128$ resolutions presented in the following paragraph where the effect of model size is also observed.

**Scalability of CBMs Compared to DBMs.** To evaluate the scalability of CBMs, we assess their performance across multiple image resolutions in comparison to DBMs. Specifically, we train separate CBM and DBM models from scratch on the Face2Comics dataset at $64 \times 64$, $128 \times 128$, and the primary $256 \times 256$ resolution, as summarized in Table 5. We compare CBMs against DBMs using HH and DBIM samplers. To accommodate the increase in resolution, we increase the model capacity to meet dimensional requirements and ensure sufficient representational power for handling more detailed inputs. We use 101M, 139M, and 161M parameters for the $64 \times 64$, $128 \times 128$, and $256 \times 256$ models, respectively. Additional architectural details are provided in Appendix C.1.

**Limitations and Future Work.** Our work explores discrete-time CBMs, which can be affected by discretization errors, requiring careful scheduling of timesteps. Following Lu & Song (2024), investigating continuous-time frameworks for CBMs can present a promising research direction that will help to address these issues. Next, analogous to CMs, the FID performance of CBMs also plateaus with NFEs, providing diminishing returns with an increase in sampling steps. This aspect can be tackled in future research, similar to Xie et al. (2024). Further research could explore

Table 3: Quantitative results on CelebAMask-HQ $256 \times 256$ for label-to-face generation. CBMs achieves superior FID with fewer NFEs and a smaller model.

| Method | Model Size ($\downarrow$) | Training Iterations ($\downarrow$) | CelebAMask-HQ ($256\times256$) [14] | | |
|---|---|---|---|---|---|
| | | | Time ($\downarrow$) | NFE ($\downarrow$) | FID ($\downarrow$) |
| *GANs & Other Diffusion-based Models:* | | | | | |
| Pix2Pix [12] | - | - | - | 1 | 56.99 |
| CycleGAN [38] | - | - | - | 1 | 78.23 |
| BBDM [15] | 534 Mil. | - | - | 200 | 21.35 |
| *Diffusion Bridge Models:* | | | | | |
| Hybrid Heun [37] | 534 Mil. | $15.36\times10^6$ | 409.84 | 119 | 97.75 |
| DBIM [36] | 534 Mil. | $15.36\times10^6$ | 20.64 | 6 | 44.92 |
| | 534 Mil | $15.36\times10^6$ | 104.24 | 30 | 18.99 |
| *Consistency-based Models:* | | | | | |
| CBM (Ours) | 350 Mil. | $15.36\times10^6$ | 8.10 | 4 | 18.41 |
| | 350 Mil. | $15.36\times10^6$ | 16.19 | 6 | **13.14** |

Table 4: Quantitative results on stylization on Face2Comics $256 \times 256$. CBMs achieve superior FID with fewer NFEs and a smaller model. Notably, our CBM reaches an FID of 3.64 with just 6 NFEs, outperforming HH and DBIM samplers by a significant margin.

| Method | Model Size | Face2Comics ($256\times256$) [31] | | |
|---|---|---|---|---|
| | | Time ($\downarrow$) | NFE ($\downarrow$) | FID ($\downarrow$) |
| *Diffusion Bridge Models:* | | | | |
| Hybrid Heun [37] | 534 Mil. | 24.57 | 20 | 52.39 |
| DBIM [36] | 534 Mil. | 9.28 | 4 | 14.56 |
| | 534 Mil. | 25.64 | 6 | 13.29 |
| *Consistency Bridge Models:* | | | | |
| CBM (Ours) | 161 Mil. | 6.30 | 4 | 14.06 |
| | 161 Mil. | 17.38 | 6 | **3.64** |

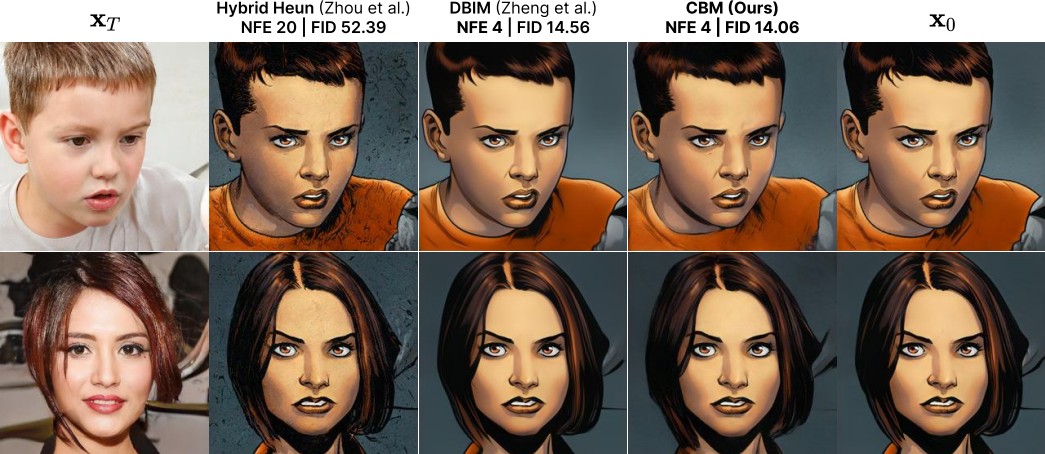

Figure 8: Qualitative comparison between images generated by a DBM using the HH and DBIM samplers, and a CBM with the Multistep Bridge sampler. When the DBM is sampled for 7 and 14 NFEs, the generated images are noisy and low-quality, whereas our CBM produces high-quality images. Numerical analyses presented in Table 4 support these results. Performed on Face2Comics $256 \times 256$ (Sxela, 2021); $\mathbf{x}_T$ are Human Faces, and $\mathbf{x}_0$ are Comics.

extending CBMs to complex I2I tasks, such as text-based conditional generation (Luo et al., 2023), video synthesis, and Video-Language-Action Models.

## 5 CONCLUSION

In conclusion, our research advances the field of I2I translation by introducing Consistency Bridge Models (CBMs), which unify the iterative refinement capabilities of Diffusion Bridge Models (DBMs) with the sampling efficiency of Consistency Models (CMs) (Song et al., 2023). CBMs emerge as a compelling alternative to traditional diffusion-based approaches such as DDBM (Zhou et al., 2023), BBDM (Li et al., 2023), and $I^2$SB (Liu et al., 2023), offering fast, few-step I2I translation without compromising output quality. By addressing key limitations in both speed and fidelity, CBMs pave the way for scalable, high-quality I2I applications across diverse domains. We believe this framework opens new avenues for future research in efficient generative modeling.

**Use of Large Language Models.** LLMs were employed exclusively for editorial refinement, without influencing research design or substantive content.

**Ethics Statement.** We affirm compliance with the ICLR Code of Ethics (https://iclr.cc/public/CodeOfEthics). This work presents CONSISTENCY BRIDGE MODELS, a generalization of consistency models that accelerates image-to-image translation compared to prior diffusion-based approaches. All experiments utilized publicly available datasets; no human subjects or private data were involved. Care was taken to avoid generating harmful, biased, or misleading content. Although generative models carry potential misuse risks, our approach enhances efficiency and fidelity without introducing new ethical concerns. We advocate responsible deployment and transparent reporting in practical applications.

**Reproducibility Statement.** We have prioritized reproducibility by detailing the CBM training and sampling algorithms, including their mathematical foundations and implementation, in the main text and appendix. All datasets are publicly accessible and properly cited. The CBM training and sampling pseudo-algorithms, evaluation protocols, and metrics are all fully described, and we intend to publish the source code with reproduction instructions.

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

# A  DIFFERENT FORMULATIONS FOR DIFFUSION MODELS

As introduced in Section 2.1, the reverse diffusion process is given by the PF ODE (Anderson, 1982; Song et al., 2020b):

$$\mathrm{d}\mathbf{x}_t = \left[ \boldsymbol{f}(\mathbf{x}_t, t) - \frac{1}{2} g(t)^2 \nabla_{\mathbf{x}_t} \log p_t(\mathbf{x}) \right] \mathrm{d}t, \tag{7}$$

where the drift and diffusion coefficients are defined as:

$$\boldsymbol{f}(\mathbf{x}_t, t) = \mathbf{x}_t \frac{\mathrm{d}}{\mathrm{d}t} \log \alpha_t, \text{ and } g(t)^2 = -2\sigma_t^2 \frac{\mathrm{d}}{\mathrm{d}t} \log \left( \frac{\alpha_t}{\sigma_t} \right),$$

where $\alpha_t := \alpha(t)$ and $\sigma_t := \sigma(t)$, for time $t \in [0, T]$ (where $T > 0$).

Different choices for $\alpha_t$ and $\sigma_t$ lead to distinct formulations of the diffusion process. Prior work has proposed hand-crafted schedules to instantiate specific types of diffusion models, including the *variance-preserving* (VP) (Song et al., 2020b; Zhou et al., 2023), *variance-exploding* (VE) (Karras et al., 2022), and *TrigFlow* (Lu & Song, 2024) formulations. We summarize and compare the designs of these formulations in Table 6.

Table 6: Design choices for widely-used diffusion formulations.

| Formulation | $\alpha_t$ | $\sigma_t$ | $\boldsymbol{f}(\mathbf{x}_t, t)$ | $g(t)^2$ | $\mathrm{SNR}_t = \alpha_t^2/\sigma_t^2$ | Domain of $t$ |
|---|---|---|---|---|---|---|
| VP [27; 37] | $e^{-(0.5t^2+0.05t)}$ | $\sqrt{1 - e^{-(t^2+0.1t)}}$ | $-(t+0.05)\mathbf{x}_t$ | $2t+0.1$ | $1/(e^{(t^2+0.1t)}-1)$ | $[0.0001, 1]$ |
| VE [13] | $1$ | $t$ | $\mathbf{0}$ | $2t$ | $1/t^2$ | $[0.002, 80]$ |
| TrigFlow [18] | $\cos(t)$ | $\sin(t)$ | $-\tan(t)\,\mathbf{x}_t$ | $2\tan(t)$ | $\cot^2(t)$ | $[0, \pi/2]$ |

# B  PRECONDITIONING AND PARAMETRIZATION OF CBMs

As discussed in Section 3.1, we parametrize CBM $\boldsymbol{f}_{\boldsymbol{\theta}}$ as a combination of input $\mathbf{x}_t$ and network $\boldsymbol{F}_{\boldsymbol{\theta}}$ for stable training as:

$$\boldsymbol{f}_{\boldsymbol{\theta}}(\mathbf{x}_t, t, \mathbf{x}_T, T) = c_{\mathrm{skip}}(t, T)\, \mathbf{x}_t + c_{\mathrm{out}}(t, T)\, \boldsymbol{F}_{\boldsymbol{\theta}}(c_{\mathrm{in}}(t, T)\, \mathbf{x}_t, c_{\mathrm{noise}}(t), \mathbf{x}_T),$$

where $c_{\mathrm{in}}(t, T)$, $c_{\mathrm{noise}}(t)$, $c_{\mathrm{skip}}(t, T)$, and $c_{\mathrm{out}}(t, T)$ are time-dependent differentiable scaling functions. Works such as Song et al. (2023) and Karras et al. (2022) also follow such formulations to improve the training stability.

Denote $\sigma_{\mathrm{data}}^2$, $\sigma_{\mathrm{prior}}^2$, and $\sigma_{\mathrm{cov}}$ as the variance of $\mathbf{x}_0$, variance of $\mathbf{x}_T$, and the covariance between the two, respectively. Further, let $\mathrm{SNR}_t := \alpha_t^2/\sigma_t^2$, the signal-to-noise ratio at time $t$. Let $a(t, T) := \frac{\alpha_t}{\alpha_T} \cdot \frac{\mathrm{SNR}_T}{\mathrm{SNR}_t}$, $b(t, T) := \alpha_t \left(1 - \frac{\mathrm{SNR}_T}{\mathrm{SNR}_t}\right)$, and $c(t, T) = \sigma_t \sqrt{1 - \frac{\mathrm{SNR}_T}{\mathrm{SNR}_t}}$. We detail the formulations for $c_{\mathrm{in}}(t, T)$, $c_{\mathrm{skip}}(t, T)$, and $c_{\mathrm{out}}(t, T)$ in Table 7. Lastly, we define the noise scaling, $c_{\mathrm{noise}}(t) := 1000\, t$.

Table 7: Preconditionings for CBM Parametrization. Let $a_t := a(t, T)$, $b_t := b(t, T)$, and $c_t := c(t, T)$.

| $c_{\text{in}}(t, T)$ | $c_{\text{skip}}(t, T)$ | $c_{\text{out}}(t, T)$ |
|---|---|---|
| $\left( a_t^2 \sigma_{\text{prior}}^2 + b_t^2 \sigma_{\text{data}} + 2 a_t b_t \sigma_{\text{cov}} + c_t^2 \right)^{-1/2}$ | $\left[ a_t^2 \left( \sigma_{\text{data}}^2 \sigma_{\text{prior}}^2 - \sigma_{\text{cov}}^2 \right) + \sigma_{\text{data}}^2 c_t^2 \right]^{1/2} \cdot c_{\text{in}}(t, T)$ | $\left( b_t \sigma_{\text{data}}^2 + a_t \sigma_{\text{cov}} \right) \cdot c_{\text{in}}^2(t, T)$ |

## C EXPERIMENT DETAILS

### C.1 TRAINING DETAILS

We provide thorough details for the training procedures of DBMs and CBMs in Table 8 and Table 9, respectively.

Table 8: Training details for DBMs for various I2I Translation tasks.

| Dataset | Edges2Handbags (E2H) [12] | DIODE [33] | Conditional Inpainting on ImageNet [3] | CelebAMask-HQ [14] | Face2Comics [31] |
|---|---|---|---|---|---|
| **Hyperparameters and Training Details** | | | | | |
| Bridge Formulation | VP | VP | I²SB [16] | VP | VP |
| Noise Conditioning, $c_{\text{noise}}$ | $250 \ln t$ | $250 \ln t$ | $1000\,t$ | $250 \ln t$ | $250 \ln t$ |
| Learning Rate | 1e-4 | 1e-4 | 1e-4 | 2e-4 | 2e-4 |
| EMA Rate | 0.9999 | 0.9999 | 0.9999 | 0.9993 | 0.9993 |
| Noise Discretization Schedule | Karras | Karras | Karras | Karras | Karras |
| Noise Discretization Steps | 40 | 40 | 40 | 40 | 40 |
| Batch Size | 256 | 64 | 256 | 64 | 64 |
| Total Training Iterations | 400K | 400K | 2380K | 120K | 120K |
| Number and Type of GPUs | 4 A100 | 4 A100 | 8 A800 (& V100) | 8 A6000 | 8 A6000 |
| **Model Details** | | | | | |
| Model Channels | 192 | 256 | 256 | 256 | 256 |
| Dropout | 10% | 10% | 10% | 10% | 10% |
| Time Embedding | Cosine | Cosine | Cosine | Cosine | Cosine |
| Channel Multiplier | (1, 2, 3, 4) | (1, 1, 2, 2, 4, 4) | (1, 1, 2, 2, 4, 4) | (1, 1, 2, 2, 4, 4) | (1, 1, 2, 2, 4, 4) |
| Number of Residual Layers | 3 | 2 | 2 | 2 | 2 |
| Attention Resolutions | (8, 16, 32) | (8, 16, 32) | (8, 16, 32) | (8, 16, 32) | (8, 16, 32) |
| Model Capacity (Mparams) | 284 | 534 | 534 | 534 | 534 |

Table 9: Training details for CBMs for various I2I Translation tasks.

| Dataset | Edges2Handbags (E2H) [12] | DIODE [33] | Conditional Inpainting on ImageNet [3] | CelebAMask-HQ [14] | Face2Comics [31] |
|---|---|---|---|---|---|
| **Hyperparameters and Training Details** | | | | | |
| Bridge Formulation | VE | VE | VE | VE | VE |
| Noise Conditioning, $c_{\text{noise}}$ | $250 \ln t$ | $250 \ln t$ | $1000\,t$ | $250 \ln t$ | $250 \ln t$ |
| Learning Rate | 1e-4 | 1e-4 | 2e-4 | 2e-4 | 2e-4 |
| EMA Rate | 0.9999 | 0.9999 | 0.999 & 0.9993 | 0.9993 | 0.9993 |
| Noise Discretization Schedule | Karras | Karras | Karras | Karras | Karras |
| Noise Discretization Steps | 40 | 40 | 40 | 40 | 40 |
| Batch Size | 256 | 64 | 160 | 64 | 64 |
| Total Training Iterations | 400K | 400K | 168K | 120K | 120K |
| Number and Type of GPUs | 4 A100 | 4 A100 | 4 H200 | 8 A6000 | 8 A6000 |
| **Model Details** | | | | | |
| Model Channels | 192 | 256 | 256 | 256 | 256 |
| Dropout | 10% | 10% | 0% | 10% | 10% |
| Time Embedding | Cosine | Cosine | Cosine | Cosine | Cosine |
| Channel Multiplier | (1, 2, 3, 4) | (1, 1, 2, 2, 4, 4) | (0.5, 0.5, 1, 2, 3, 4) | (1, 1, 2, 2, 4, 4) | (1, 1, 2, 2, 4, 4) |
| Number of Residual Layers | 3 | 2 | 2 | 2 | 2 |
| Attention Resolutions | (8, 16, 32) | (8, 16, 32) | (8, 16, 32) | (8, 16, 32) | (8, 16, 32) |
| Model Capacity (Mparams) | 284 | 534 | 398 | 534 | 534 |

## D EXTENDED RESULTS

We present more qualitative results on Conditional Inpainting ImageNet $256 \times 256$ (Deng et al., 2009) in Figure 10. We train a CBM from scratch (described in Table 9). We use the DBM model weights provided by Zheng et al. (2024), which is a heavily finetuned version of an N2I generating DPM on ImageNet $256 \times 256$ that was published by Dhariwal & Nichol (2021).

We observe that CBMs achieve an FID of 5.61 in just 4 NFE, surpassing DBM with DBIM sampler's 17.85 FID in the same NFEs. Note that compared to CBM, the DBM is trained for considerably longer; the CBM was trained only for 26.88M images, whereas the DBM is trained for 609.28M images in total (506.88M for DPM, and 102.4M for additional finetuning), taking relatively over $21\times$ longer. This further shows the training efficiency of CBMs compared to DBMs.

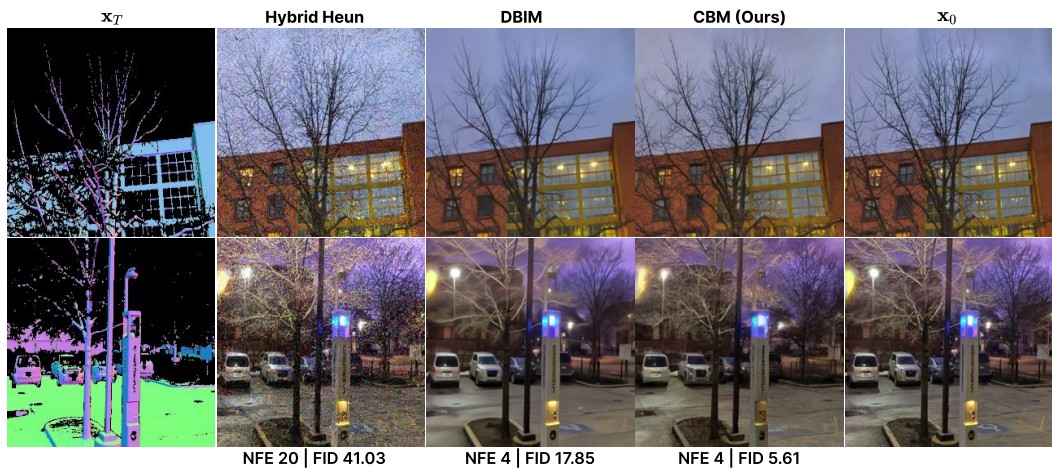

Figure 10: Qualitative comparison on the DIODE dataset (Vasiljevic et al., 2019) at $256 \times 256$ resolution. Our CBMs with Multistep Bridge Sampler generate higher-quality images with fewer NFEs when compared to DBMs with HH (Zhou et al., 2023) and DBIM (Zheng et al., 2024) (Table 2). $\mathbf{x}_T$ are Surface Normal Maps, and $\mathbf{x}_0$ are Images.

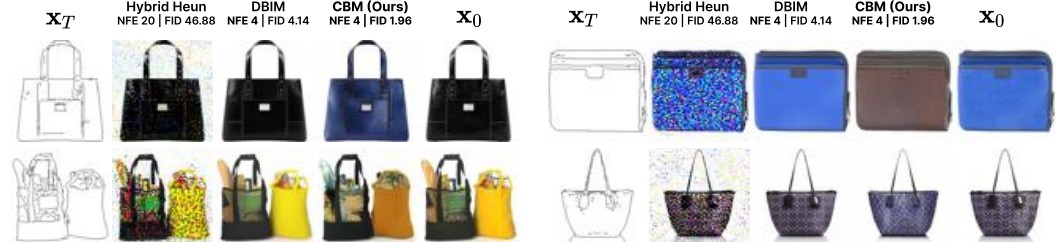

Figure 11: Qualitative comparison on Edges2Handbags $64 \times 64$ (Isola et al., 2017). CBMs produce superior visual outputs with reduced numerical function evaluations relative to the HH (Zhou et al., 2023) and DBIM (Zheng et al., 2024) approaches employed in traditional DBMs (Table 2). $\mathbf{x}_T$ are Edges, and $\mathbf{x}_0$ are Images. Note that when sampling CBMs (Algorithm 2), every intermediate $\mathbf{x}_{t_i}$ contains randomly added noise that results in the final output image looking slightly different from the ground truth $\mathbf{x}_0$ image. Regardless, they maintain high-level of realism and image quality, as seen in the quantitative results.

We present additional results for E2H $64 \times 64$ in Figure 11, and CelebAMask-HQ $256 \times 256$ in Figure 12. Once again, CBMs achieve better image quality and FIDs in fewer NFEs compared to DBMs with HH or DBIM samplers.

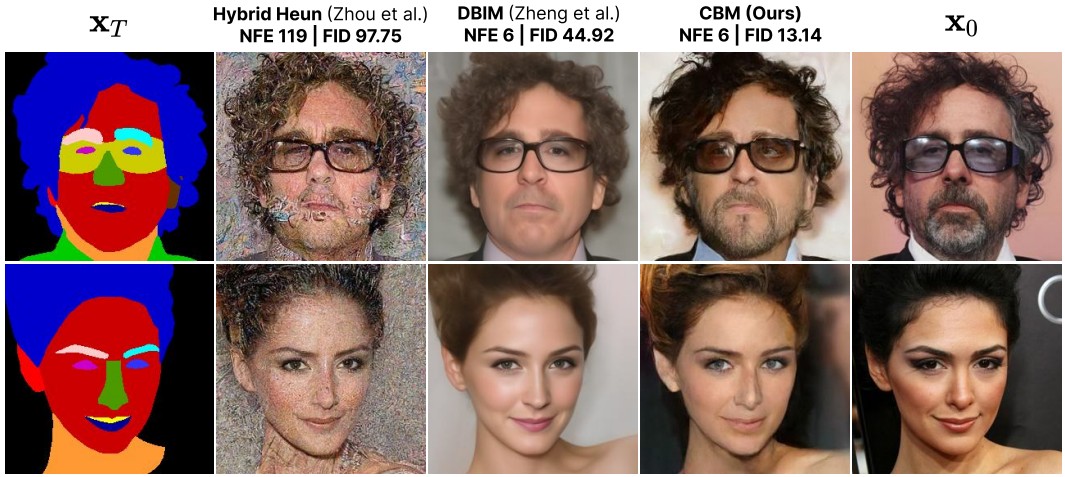

Figure 12: Qualitative comparison of CBM to other baselines on CelebAMask-HQ $256 \times 256$.

