# OpenReview forum: "Generalizing Consistency Models for Fast Image-to-Image Translation"
_ICLR.cc/2026/Conference — ICLR 2026 Conference Withdrawn Submission_

### Official Review · Reviewer_Mkx3 · 2025-10-16

**Soundness:** 2
**Presentation:** 2
**Contribution:** 2
**Rating:** 4
**Confidence:** 3

**Summary:**

The paper proposed consistency bridge models (CBM), which are trained with the consistency objective for bridge image-to-image translation models. The key contribution of the paper lies in the formulation of consistency training framework in the Equation 6, which generalizes the consistency training loss from noise-to-image diffusion models to image-to-image bridge models. The evaluation of the proposed CBM method includes edges2handbags, DIODE-Outdoor, face2comic, and CelebAMaskHQ image-to-image translation problems. CBM is compared with closely related diffusion bridge models of DBIM, DDBM, and the distilled version of DDBM, namely CDBM. These image-to-image translation models are evaluated using FID for image quality, inference time and NFE for inference efficiency, training time for training efficiency, and model size for computational budget. The results of CBM show its advantages against DBIM and DDBM in terms of quality and CDBM in terms of training and inference efficiency.

**Strengths:**

1) The method proposed to train diffusion bridge image-to-image translation models with consistency losses with small NFE, which does not require additional finetuning used for distillation diffusion bridge models with consistency loss in CDBM.
2) CBM is evaluated and compared with DDBM and DBIM methods on 4 image-to-image translation problems, which supports its efficiency compared with existing open-sourced diffusion bridge models.
3) Figure 6 shows advantage of the proposed CBM in terms of training time compared with distillation-based method of CDBM with consistency loss on 2 image-to-image translation problems of DIODE and edges2handbags.

**Weaknesses:**

1) The work does not provide any theoretical analysis of the proposed consistency bridge loss to support the correctness of the CBM. Such analysis is given in Theorem 2 in Consistency Models for consistency training loss for noise-to-image diffusion models without distillation or finetuning and in Proposition 3.2 and 3.3 for distillation and finetuning of diffusion bridge models in CDBM.
2) The presented comparison with CDBM is not enough to state the efficiency and novelty of the proposed CBM in relation to existing efficient diffusion bridge image-to-image translation models. Firstly, according to Table 2, CDBM achieves better FID with less NFE compared with CBM on both datasets, but requires longer training of bigger model size. It is not clear what is more important for this gap - training time or the model size - and can CBM beats CDBM when the computational budget of CDBM is similar to CBM in terms of model size and training time. Secondly, as noted in IBMD paper, Section 5.2, the evaluation setup for edges2handbags and DIODE-Outdoor image-to-image translation problems, which was used in DDBM and DBIM, provide the FID metric on training set, but testing sets from those models are too small for FID computation. It can lead to the problem of poor generalization for the trained models. The comparison between CDBM and CBM for other image-to-image translation problems is necessary due to this observation.
3) To evaluate the image quality of translated images, the method used only the FID metric. Both DBIM and DDBM methods computed other metrics as well (LPIPS, MSE, IS) to evaluate the bridge models on the problems of DIODE and edges2handbags. Additional image quality metrics are important to evaluate not only the unpaired realism of translated images, but their correspondence to the reference images as well.
4) The experimental study of the CBM method lacks of ablation studies. In lines 206-207, authors write that used pseudo-Huber or MSE distance functions for $d$, but it is now clear which distance was actually used in Algorithm 1. Also, the study on NFE $N$ in Algorithm 2 considers only NFE=4 and NFE=6. It is not clear what is the drop quality of CBM for NFE=2 as was used in CDBM and can CBM achieve stable results with the increase of NFE. Consistency Training models also can be evaluated with NFE=1, as shown in the Table 1 and 2 of the corresponding paper.

References:

Consistency Models. ICML-2023.
IBMD - Inverse Bridge Matching Distillation. ICML-2025.

**Questions:**

1) According to lines 694-695, the Figure 10 should present the qualitative and quantitative results of CBM for the inpainting problem on ImageNet. However, according to the caption in lines 717-718, Figure 10 shows the results on the DIODE dataset. Is it true, that numbers in Figure 10 correspond to the inpainting problem on ImageNet, but visual results for this problems are missed?
2) Can you provide additional metrics for the results of diffusion bridge models using LPIPS and MSE following DDBM and DBIM?
3) Can you comment why there is difference between model sizes of CDBM and CBM on DIODE-Outdoor dataset in Table 2? According to Table 8 and 9 models sizes of CBM and CDBM should be the same 534 million of parameters.
4) Can you comment about the CDBM quality drop if the total training time of the bridge model and its distillation will be similar to the total training time of CDBM and DDBM and how this quality drop relates to the quality drop of CBM when using NFE=2, as in CDBM, instead of NFE=4 or NFE=6?
5) Can you also comment on the inference time in seconds of CDBM and CBM since model sizes of these models might be different?
6) Can you comment how the choice of distance metric $d$ affects the results? For example, Consistency Model paper also considered LPIPS for similarity $d$d.
7) According to lines 695-696 authors write: "We train a CBM from scratch (described in Table 9). We use the DBM model weights provided by Zheng et al. (2024), which is a heavily finetuned version of an N2I generating DPM on ImageNet". Is it correct that in your model all weights were initialized randomly before the training?
8) Distillation methods of IBMD and CDBM for diffusion bridge models distilled the DDBM architecture for the same inpainting problem on ImageNet. Can you comment what if the DDBM model was distilled with CDBM or IBMD methods for the inpainting problem on ImageNet in such a way, so that the total number of training images is comparable with CBM?

References:

IBMD - Inverse Bridge Matching Distillation. ICML-2025.

---

### Official Review · Reviewer_4u3Z · 2025-10-17

**Soundness:** 3
**Presentation:** 2
**Contribution:** 1
**Rating:** 2
**Confidence:** 3

**Summary:**

This paper introduces Consistency Bridge Models (CBMs) for fast and high-quality image to image translation. CBMs do not need pretrained models.

**Strengths:**

(i) The method achieves a good balance between sampling speed and performance, while using fewer training parameters.

(ii) The authors test their method on more tasks than past works [1,2,3], like label to face generation on CelebA and image stylization on Face2Comics.

[1] https://arxiv.org/abs/2309.16948 Denoising Diffusion Bridge Models

[2] https://arxiv.org/abs/2405.15885 Diffusion Bridge Implicit Models

[3] https://arxiv.org/abs/2410.22637 Consistency Diffusion Bridge Models

**Weaknesses:**

(i) Main weakness is the novelty claim made by the authors. They write: “*To the best of our knowledge, this is the first attempt to generalize the consistency framework to achieve I2I translation*” (line 160). But similar work has already been done in CDBM [3], which also applies consistency models to image to image tasks. The main difference seems to be that this paper trains models from scratch, while CDBM fine tunes pretrained DDBM [1]. If this is the only difference, the novelty claim may not be fully correct. The authors should better explain what is new in their method compared to [3].

(ii) The authors state that comparing with CDBM is unfair in terms of performance metrics since it fine tunes a pretrained DDBM, while their method trains from scratch. However, they report CDBM’s training time as the sum of DDBM pretraining and consistency fine tuning. This may be misleading, as CDBM could be faster if trained from scratch with consistency loss only. A fair comparison would require aligning training setups or reporting CDBM's scratch training time.

**Questions:**

(i) Could the authors clearly explain the differences between their method and CDBM [3], beyond the fact that CDBM uses a pretrained DDBM model? A more detailed comparison would help clarify the novelty and contributions of this work.

(ii) I suggest the authors include comparisons with more recent state-of-the-art I2I methods, such as IBMD [4], to better position their approach within the current literature.

(iii) There appear to be several issues in the text that should be addressed:
   - In Equation (2), it would be more accurate to include both `stopgrad` and EMA in the loss formulation.
   - In line 125, it should state "from $x_T$ to $x_0$" rather than "from $x_0$ to $x_T$".
   - In the algorithm (line 224), the notation should be $t_{i+1}$, not $t_{i-1}$.

[4] https://arxiv.org/abs/2502.01362 Inverse Bridge Matching Distillation

---

### Official Review · Reviewer_hEJU · 2025-10-31

**Soundness:** 2
**Presentation:** 3
**Contribution:** 1
**Rating:** 2
**Confidence:** 4

**Summary:**

This paper proposes a method for one-step image-to-image translation by training the ODE path of a DDBM (Diffusion-Denoising Bridge Model) through a consistency training framework. The authors not only apply consistency training directly to the DDBM ODE, but also extend the consistency model’s multi-step sampling scheme, enabling one-step translation. Experiments across various domains and resolutions (64×64 to 256×256) demonstrate that the method can be trained from scratch without requiring a separately pretrained diffusion model. The results show comparable or slightly lower FID performance relative to competing approaches, while requiring less total training time.

**Strengths:**

1. The problem of image-to-image translation remains meaningful and relevant, and the focus on improving few-step methods in this setting is well-justified.
2. The proposed framework is a reasonable and direct extension of applying consistency training to the DDBM ODE. The methodology is intuitive, clearly explained, and should be easy to understand for readers familiar with both DDBM and consistency models.
3. The paper is generally clearly written and easy to follow.

**Weaknesses:**

1. The key concern is novelty. It is difficult to distinguish this work from prior work on CDBM [1]. While the authors emphasize that CDBM depends on a pretrained model and their approach trains from scratch (L254), the CDBM paper also presents a from-scratch consistency training variant. Moreover, the actual training objectives appear nearly identical. This raises significant concerns regarding the originality of the contribution.
2. The multi-step sampling strategy appears to follow straightforwardly from applying a consistency model to the DDBM ODE, and therefore does not seem to represent an additional conceptual contribution.
3. The experimental results are limited and do not show clear improvements. Compared to CDBM, the proposed model achieves worse FID and requires ~3× more NFEs at inference. Although the paper claims advantages in training time, the difference is small, and, as noted above, CDBM also supports from-scratch training—so the comparison is not fully fair. Additionally, evaluation relies solely on FID, which is insufficient to establish translation quality.
4. The claim in L160 that “this is the first consistency-model framework for image-to-image translation” is incorrect and misleading. Prior works such as CDBM [1] and IBCD [2] already apply CM-based formulations to I2I settings, albeit under slightly different constraints and formulations.

[1] He, Guande, et al. "Consistency diffusion bridge models." NeurIPS (2024).

[2] Lee, Suhyeon, Kwanyoung Kim, and Jong Chul Ye. "Single-Step Bidirectional Unpaired Image Translation Using Implicit Bridge Consistency Distillation." *arXiv* (2025).

**Questions:**

1. In Algorithm 1 and L201, the training pairs ($x_0$, $x_T$) appear to be sampled independently from $p_0(x)$ and $p_T(x)$, rather than from the joint distribution $p(x_0, x_T)$. However, DDBM is trained on paired datasets, i.e., $(x_0, x_T) \sim p(x_0, x_T)$. Is the model trained on paired or unpaired data? Is this notation intentional?

---

### Official Review · Reviewer_ucGk · 2025-10-31

**Soundness:** 1
**Presentation:** 1
**Contribution:** 1
**Rating:** 2
**Confidence:** 5

**Summary:**

The paper introduced a consistency-based diffusion bridge model for fast sampling in image-to-image translation task, which is trained using consistency loss between an online model and an EMA model to predict the same clean data given two different noise level input. The paper also presented multi-step sampling procedure with consistency bridge model to improve performance.

**Strengths:**

– CBM achieves state-of-the-art qualitative and quantitative results at very low NFE on several I2I benchmarks, showing strong few-step performance.

**Weaknesses:**

– Novelty is very limited. This work is highly similar to CDBM [1], but the comparison is brief. The paper does not clearly articulate what is different or novel relative to CDBM, especially CDBM already proposed training consistency from scratch.

– The paper argued that CDBM are constrained by requiring a pretrained DBM, which is not true. CDBM can also train from scratch, using pretrained DBM only provides a good initialization.

– There are issues in writing (background of consistency models, diffusion bridges) and, more importantly, in the design of the consistency objective, making the results unconvincing. The SOTA gains over baselines are not thoroughly explained.

– A notable flaw: training consistency from scratch requires the two samples at different noise scales to share the same score $\nabla\log p\left(x_{t}\mid x_{T}\right)$, which which can be estimated via $\nabla\log p\left(x_{t}\mid x_{0},x_{T}\right)$ as in [1] (i.e. the same $z$). However, the paper samples $x_{t_{i}}$ and $x_{t_{i-1}}$ independently given $x_{0}$ and $x_{T}$. This does not ensure they share the same score, so the loss no longer matches the intended score-coupling in consistency training.

– The paper is incomplete as Appendix D is missing. The efficiency comparison is unclear: CBM appears to employ a pretrained DBM checkpoint (rather than retraining from scratch) for noise-to-image generation; thus the claim of higher efficiency is not validated.

[1] He, Guande, et al. "Consistency diffusion bridge models." NeurIPS 2024

**Questions:**

– What is the methodological difference between CBM and the training variants of CDBM (including teacher-free training)?

– Why does CBM require less training duration than CDBM when the methods are very similar?

– Please compare CBM against both CDBM variants (fine-tuning and training from scratch) for fairness.

– What is the dataset setup? If train and evaluation sets overlap, results may reflect memorization; please clarify splits, report test-set metrics and diversity score to show that CBM is not overfitted to training dataset.

– Why does CBM outperform baselines? Provide analysis/ablations isolating the factors that boosting the performance

---

### Note · Authors · 2025-11-13

I have read and agree with the venue's withdrawal policy on behalf of myself and my co-authors.